# Tunable Fiber Laser with High Tuning Resolution in C-band Based on Echelle Grating and DMD Chip

**DOI:** 10.3390/mi10010037

**Published:** 2019-01-08

**Authors:** Jinliang Li, Xiao Chen, Dezheng Dai, Yunshu Gao, Min Lv, Genxiang Chen

**Affiliations:** 1College of Science, MINZU University of China, Beijing 100081, China; jinliangft@163.com (J.L.); daidezheng@126.com (D.D.); lvmin62589149@163.com (M.L.); 2School of Electronic and Information Engineering, Beijing Jiaotong University, Beijing 100044, China; gaoyunshu@126.com

**Keywords:** tunable fiber laser, echelle grating, DMD chip

## Abstract

The tunable fiber laser with high tuning resolution in the C-band is proposed and demonstrated based on a digital micromirror device (DMD) chip and an echelle grating. The laser employs a DMD as a programmable wavelength filter and an echelle grating with high-resolution features to design a cross-dispersion optical path to achieve high-precision tuning. Experimental results show that wavelength channels with 3 dB-linewidth less than 0.02 nm can be tuned flexibly in the C-band and the wavelength tuning resolution is as small as 0.036 nm. The output power fluctuation is better than 0.07 dB, and the wavelength shift is below 0.013 nm in 1 h at room temperature.

## 1. Introduction

Tunable lasers as a powerful tool have been widely applied in spectroscopy, photochemistry, biomedicine, and optical communications for decades. For example, in dense wavelength division multiplexing (DWDM) optical communication, tunable lasers can not only replace multiple fixed-wavelength lasers to save the operation cost but also realize the remote dynamic allocation of networks resources. The number of wavelength channels in C-band determines the information transmission capacity in networks. Therefore, how to improve narrow-linewidth channels with a high tuning accuracy from laser sources has been receiving an increasing amount of attention from researchers and network service vendors.

To date, various technologies have been proposed and implemented to realize tunable filters in laser sources, including fiber Bragg grating (FBG), Fabry–Perot (F–P) cavity, acousto-optics, interferometer, liquid crystal on silicon (LCoS), etc. FBG can be tuned easily through either heating or applying strain along the device. For example, it is reported that FBG can achieve 0.2 nm/V tuning accuracy from 1555–1565 nm driven by direct current (DC) voltage of multilayer piezoelectric transducers [1]. However, FBG-based tunable lasers are affected by the environment fluctuation, resulting in a high packaging cost and limited tuning range. The fiber-optic self-seeding F–P cavity achieves a wide range of single longitudinal modes tuning from 1153.75 to 1560.95 nm with a tuning step of 1.38 nm [2]. Avanaki et al. investigate a fiber Fabry–Pérot tunable filter using a well-established optimization method, simulated annealing (SA), to achieve maximum amplitude for the Fourier transformed peaks of the photodetected interferometric signal [3]. Furthermore, Y. Ding implemented a small-scale tuning with the accuracy of approximately 0.6 nm by using micro-ring Mach–Zehnder interferometers [4]. These technologies generally need additional matching devices, like an F–P laser, saturable absorber-based filters, which makes them complex and expensive to commercialize. Nowadays, a LCoS spatial light modulator as a programmable filter produced by Very-Large-Scale-Integrated (VLSI) technology has been applied to laser systems [5]. A digital micromirror device (DMD), another Opto-VLSI processor has also been attempted in a non-projection field. In 2006, Chuang and Lo proposed a spectral synthesis method with a spectral tuning accuracy of 0.076 nm/pixel in the C-band based on a DMD chip [6]. In 2009, W. Shin used the DMD-based tunable laser system as light sources for the optical time domain reflectometry, with a tuning range of 1525–1562 nm, and an improved laser tuning accuracy of 0.1 nm [7]. Our research group also reported a multi-wavelength tunable fiber laser based on a DMD chip with a step of 0.055 nm [8].

Echelle gratings are a special type of blazed gratings featured by a large blazing angle of grooves and often operate at high diffraction orders to obtain high dispersion. They are different to conventional gratings [9,10,11,12]. An echelle grating splits the radiant energy into a multitude of diffraction orders that overlap in the narrow interval of the grating diffraction angle. Therefore, in practical application, an additional order separator like the prism or grating, whose dispersion direction is perpendicular to that of an echelle grating are inserted to separate the overlapping orders. By focusing the two-fold dispersed radiation, a two-dimensional spectrum is produced, thus achieving an applicable high-resolution spectrum. So far, echelle gratings are mainly applied in ultraviolet and visible high-resolution spectrometers [10,11].

In this work, we first apply an echelle grating into a DMD-based tunable laser to realize the high tuning resolution in C-band. The echelle-based tunable fiber laser is designed for a cross-dispersion structure of a closed-loop fiber system. The laser wavelength was tuned in the range of 1540–1560 nm with a tuning step of 36 pm. The 3dB-linewidth of the signals was less than 0.02 nm. The side mode suppression ratio (SMSR) reaches 40 dB, and the maximum output power was 7.5 dBm.

## 2. Echelle Grating and System Design

The spectral order of an echelle grating is the result of mutual modulation of multi-slit interference and single-slit diffraction. The echelle equation is expressed as:(1)mλ=d(sinα+sinβ)cosγ
where *m*, *λ*, and *d* are the diffraction order, wavelength, and grating constant, respectively. *α*, *β* and *γ* are the incident angle, corresponding diffraction angle, and off-axis angle. As shown in Figure 1a, *θ*_B_ is the blaze angle of an echelle grating and *θ* is the incident angle to the facet. So, the relation of angles is written as:(2)$$\alpha =\theta_{B}+\theta ,\;\beta =\theta_{B}-\theta$$

Substituting Equation (2) into (1), the diffraction of an echelle grating is characterized as follows:(3)$$m\lambda =2d\sin\theta_{B}\cos\theta \cos\gamma$$

An echelle grating has the maximum diffraction efficiency only when the Littrow condition is satisfied, that is the incidence is at the blaze angle. On both sides of the blaze angle, the diffraction efficiency of a grating decreases rapidly as *θ* increases. However, the strict Littrow condition leads to the difficulty in the arrangement of the actual optical path. Therefore, a quasi-Littrow structure is usually employed with the incident ray at an off-axis angle *γ* from the principal section of a grating, as shown in Figure 1b. The condition of the quasi-Littrow configuration is:(4)θ=0, γ≠0

Therefore, the echelle grating equation under the quasi-Littrow condition is:(5)$$m\lambda =2d\sin\theta_{B}\cos\gamma$$

The free spectral range $△\lambda_{\mathrm{SFR}}$:(6)$$△\lambda_{\mathrm{SFR}}=\frac{\lambda }{m}=\frac{\lambda^{2}}{2d\sin\theta_{B}\cos\gamma }$$

The range of the dispersion angle of *m*-order:(7)$$△\theta =△\lambda_{SFR}=\frac{d\theta }{d\lambda }=\frac{2\tan\theta_{B}}{m}$$

It can be seen from Equations (5)–(7) that an echelle grating has the following features: (1) The free spectral range is small and the spectral order is seriously overlapped. Therefore, it is necessary to use auxiliary dispersion elements for cross-dispersion to obtain a two-dimensional spectrum. (2) The angular dispersion is so high that the wavelength resolution is greatly improved. (3) The dispersion angle of one order is small, and the wavelengths in the free spectral range of each stage are concentrated near the blazed order, so an echelle grating can blaze in the entire band.

In Figure 2, the two-dimensional cross-dispersion is realized by a diffraction grating and an echelle grating with main sections that are perpendicular to each other. As we know, echelle gratings are, to date, mainly applied in ultraviolet (UV) and visible (VIS) spectrometers, so most of the prisms are used as auxiliary dispersers placed before or after an echelle grating to achieve cross-dispersion. In our work, the laser operates in the C-band and the prism glass material shows strong absorption in infrared. Therefore, a diffraction grating is adopted to replace the prisms in fiber lasers.

Figure 3 demonstrates the tunable laser structure employing a DMD chip as a programmable filter in bulk optics and a fiber resonator with an erbium-doped fiber amplifier (EDFA). The lasing process in a fiber cavity is achieved by optical pumping and erbium gain. The bulk optics obtain the high-precision mode selection by an echelle grating and a 0.55″ DMD in the experiment. The detailed working principle of a 0.55″ DMD and its diffraction efficiency have been analyzed in [13]. The EDFA emits the amplified spontaneous emission spectrum (ASE) signals from 1530–1560 nm. After a 90/10 optical fiber coupler, 90% ASE light energy returns into a ring and then continues to be coupled into the bulk optics via a circulator and an optical fiber collimator. The bulk optics consists of two cylindrical lenses, a diffraction grating, an echelle grating, and a DMD chip. The fiber collimator and the 1200 line/mm diffraction grating are located at the front and the rear focal planes of lens (*f*_0_ = 100 mm), respectively. The diffraction grating and the 79 line/mm echelle grating are separated by 100 mm. In order to ensure that the echelle grating adheres to the quasi-Littrow condition, the incident beam is arranged at an off-axis angle *γ* so that the diffracted beam and the incident beam are in the same horizontal plane. The cylindrical lens 1 (*f*_1_ = 150 mm) and cylindrical lens 2 (*f*_2_ = 100 mm) are 50 mm and 100 mm from the echelle grating, respectively. Therefore, the busbars of two cylindrical lenses are perpendicular to each other, and the two dispersion directions after two gratings are collimated, respectively. The DMD is at the back focal plane of two cylindrical lenses, as shown in Figure 4. By uploading steering holograms onto the DMD controlled by remote software, any waveband of ASE spectra can be routed and coupled into the optical system along the original path, and the others are dropped out with dramatic attenuation, thereby achieving the laser longitudinal mode selection and wavelength tuning. The selected wavebands through the collimator and circulator returning into a ring cavity are amplified by EDFA, leading, after several recirculations, to high-quality single-mode laser generation.

The off-axis arrangement greatly influences the laser tuning range and accuracy. We optimize the off-axis angle *γ* of the echelle grating (79 line/mm) for the laser system. According to Equation (5), *m* = 15 and *λ* = 1550 nm are selected as the calibration blazed wavelength, and the corresponding *γ* under the quasi-Littrow condition is calculated as 18.05°. Using Zemax OpticStudio software, we design the optical system to analyze the beam distribution on the DMD surface. The simulation results illustrate the length of the two-dimensional dispersion strip is 12.2 mm in Figure 4, matching with the experimental pattern in the inset of Figure 4. The 0.55″ DMD receiving wavelength range is around 20 nm from 1540–1560 nm and is limited by the DMD size. The tuning accuracy of the laser wavelength is 0.0177 nm/pixel, in theory. Considering the used echelle grating has a wide working range from UV to 25 μm, this laser system is convenient to be extended in the 2 μm-band, which has potential applications in the biomedical domain [14,15].

## 3. Experimental Results 

When the optical loop is closed, Figure 5 shows a typical laser signal with the center of the wavelength at 1546.733 nm when the pump power is 120 mW. The power of the laser output is around 7.5 dBm, the 3 dB-linewidth is less than 0.02 nm (limited by the resolution of the YOKOGAWA spectrum analyzer, Yokogawa Test & Measurement Corporation, Tokyo, Japan), and the SMSR exceeds 40 dB.

Different holograms are loaded onto the DMD chip, each hologram corresponds to a different selected wavelength. Each selected wavelength is amplified by EDFA to achieve lasing. Figure 6 is the measured outputs of the echelle-grating-based fiber laser tuning from 1542 to 1558 nm by remotely uploading the 8 × 768 pixel-holograms at different positions along the DMD active window when the threshold pumping power is 28 mW. It demonstrates an excellent tuning capability. Notice that the range of the actual tuning wavelength is a little wider than 16 nm. The wavelength outside the tuning range requires a higher threshold power to lasing due to the off-axis angle and the influence of stray light.

Figure 7 is the fine tuning characteristics of laser outputs with the fine tuning accuracy 0.036 nm. We modulate the selected wavelength each time by moving 2-pixels on the hologram. The tuning accuracy corresponding to each pixel is related to the number of DMD pixels covered by the ASE spectrum on the surface of the DMD. Note that the tuning accuracy can be further improved by employing a DMD with a smaller pixel size, like the DLP2010NIR (Texas Instruments Incorporated, Dallas, TX, USA. Each pixel size is 5.4 μm). The shoulders on both sides of the laser spectrum may be due to self-phase modulation or other nonlinear phenomena arising from a high-level of output power [16].

Figure 8 shows the drift of wavelength (dotted line) and the fluctuation of peak power (solid line) at the pump power 40 mW during 1-h observation at the center wavelength of 1546 nm. The maximum wavelength drift is less than 0.013 nm and the maximum peak power fluctuation is 0.07 dB at room temperature. The linewidth is better than that reported in [5] (0.05 nm) and [8] (0.02 nm), and the maximum peak power fluctuation is better than that in [8] (0.25 dB). Compared with other tunable lasers with the same tuning mechanism, the laser output stability has been further improved.

Finally, due to the off-axis angles introduced into the tunable fiber laser, the aberration-like coma and astigmatism influences the tuning range and accuracy. Therefore, we will continue to optimize the optical path and reduce the stray light effect caused by an echelle grating in the follow-up work, which will be helpful to further improve the tuning property of devices. Also, loading the modulation algorithm on the DMD is an attractive solution, and our research process in the future will also consider using algorithms to further improve the performance of tunable fiber lasers.

## 4. Conclusions

The C-band tunable fiber laser based on a DMD chip and an echelle grating is proposed and demonstrated experimentally. The laser employs a DMD as a programmable wavelength filter and an echelle grating with high-resolution features to design a cross-dispersion optical path to achieve high-precision tuning. The optimal off-axis angle of an echelle grating under the quasi-Littrow condition is simulated and analyzed in detail. Experimental results show that wavelength channels are tuned in the range of 1542–1558 nm with a tuning step of 0.036 nm. The 3 dB-linewidth of the signals is less than 0.02 nm, the SMSR reaches 40 dB, and the maximum output power is 7.5 dBm. At room temperature, the output power fluctuation is better than 0.07 dB in 1 h, and the wavelength shift is below 0.013 nm.

## Figures and Tables

**Figure 1 micromachines-10-00037-f001:**
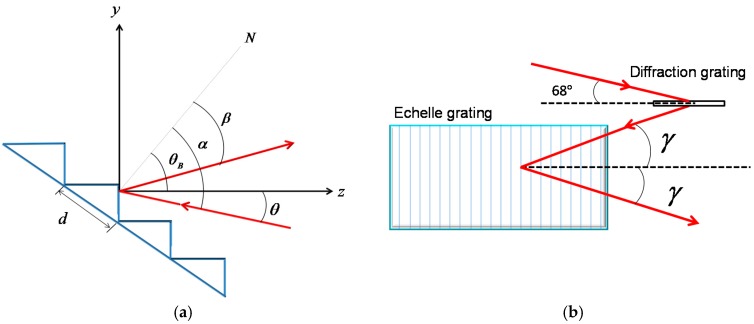
(**a**) Echelle grating working characteristics, (**b**) off-axis angles *γ* (the incident angle of diffraction grating is 68°).

**Figure 2 micromachines-10-00037-f002:**
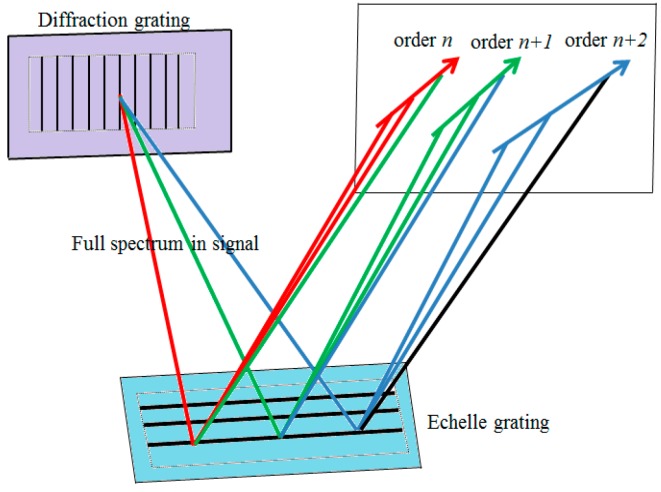
Schematic of cross-dispersion of the diffraction grating and the echelle grating.

**Figure 3 micromachines-10-00037-f003:**
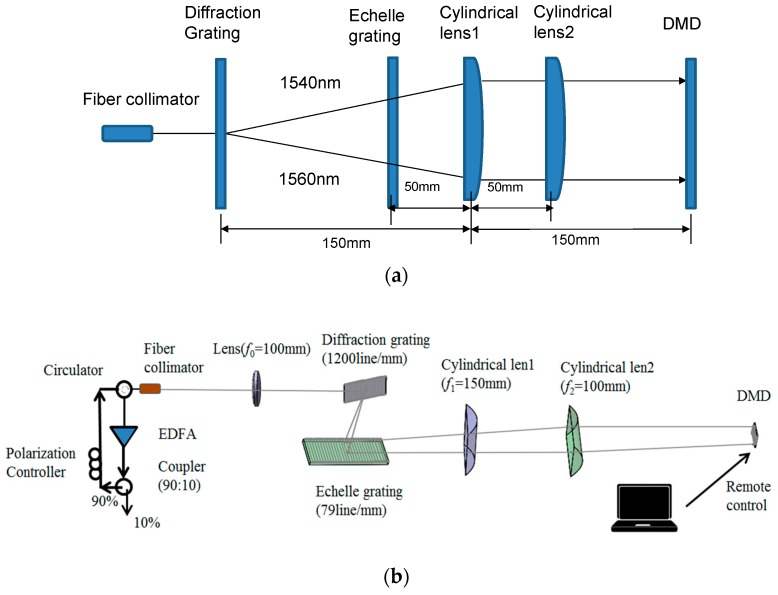
(**a**) Diagram of a cross-dispersion tunable laser system. (**b**) Layout of wavelength selective path.

**Figure 4 micromachines-10-00037-f004:**
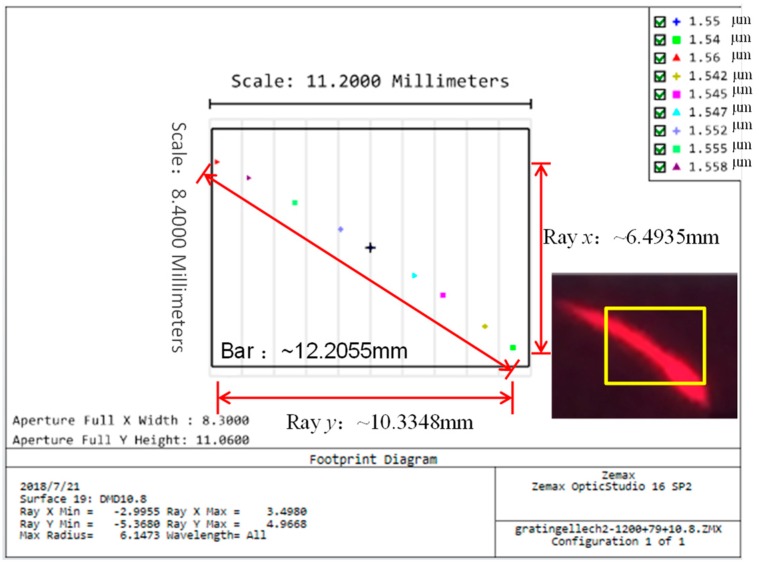
Distribution of the dispersion bar on the digital micromirror device (DMD) surface simulated by Zemax OpticStudio. The inset is the experimental pattern (rectangular box is illustrated as a DMD).

**Figure 5 micromachines-10-00037-f005:**
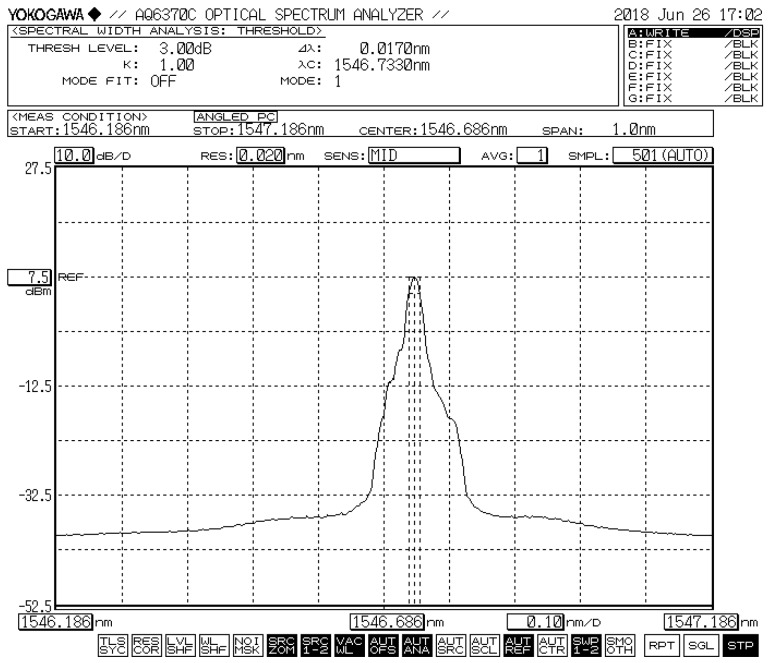
Typical laser signal with the wavelength centered at 1546.733 nm.

**Figure 6 micromachines-10-00037-f006:**
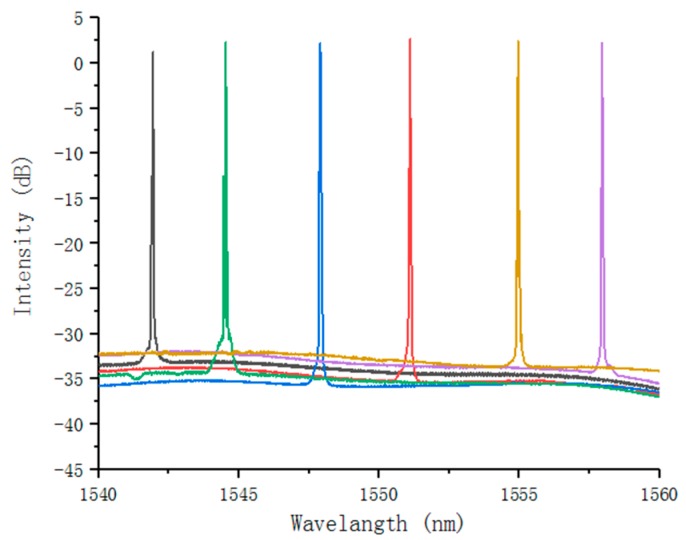
Coarse tuning characteristics of fiber laser in the range 1542–1558 nm.

**Figure 7 micromachines-10-00037-f007:**
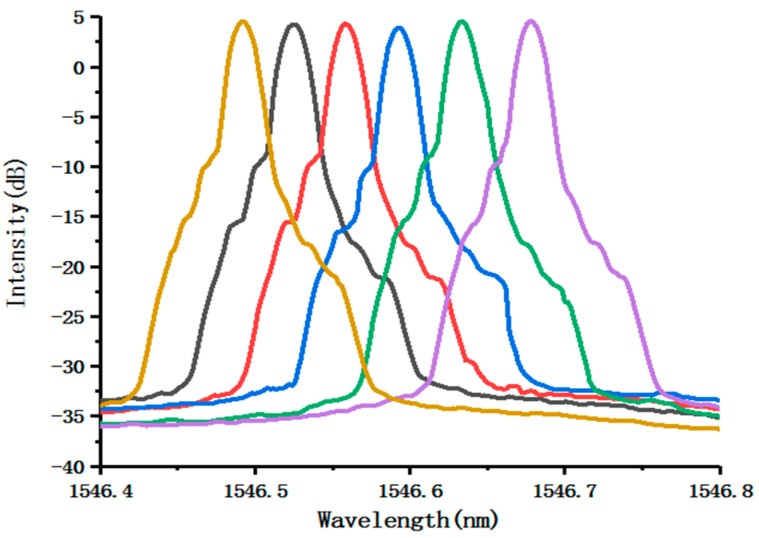
Fine tuning characteristics of the laser system from 1546.4–1546.8 nm with a tuning step of 0.036 nm.

**Figure 8 micromachines-10-00037-f008:**
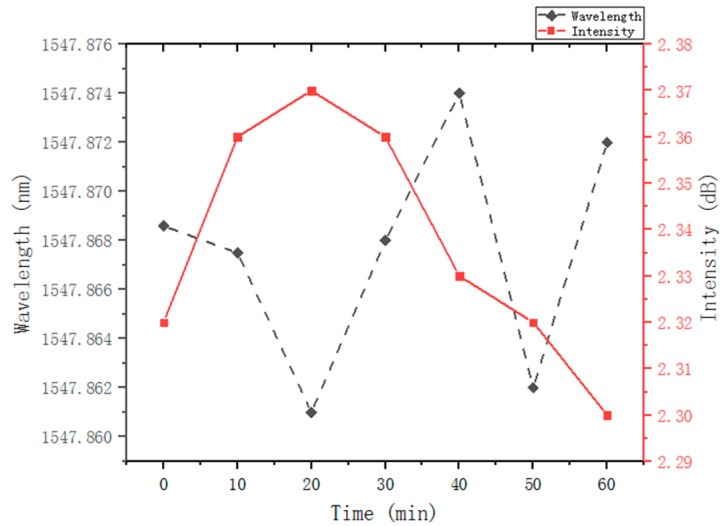
The shift of the center wavelength (dash line) and the fluctuation of laser powers (solid line) within 1 h.

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
