# Peer review of "Tunable Fiber Laser with High Tuning Resolution in C-band Based on Echelle Grating and DMD Chip"

_micromachines, 2019, doi:10.3390/mi10010037_

Round 1

Reviewer 1 Report

This manuscript introduces an approach to use echelle grating to find tune the resolution of a tunable fiber laser. The paper is well written and easy to follow. Only two minor comments:

1) Abstract, DMD was shown the first time here in abstract, needs to have the full name of DMD.

2) Line 182: Maximum wavelength drift is 0.013nm and peak power fluctuation is 0.07dB, so how does these values compared with the common tunable laser? Are they smaller or bigger?

Author Response

Reviewer 1:

The reviewer’s comment: This manuscript introduces an approach to use echelle grating to find tune the resolution of a tunable fiber laser. The paper is well written and easy to follow. Only two minor comments:

1) Abstract, DMD was shown the first time here in abstract, needs to have the full name of DMD.

Answer1: The full name of DMD is added in abstract marked yellow.

2) Line 182: Maximum wavelength drift is 0.013nm and peak power fluctuation is 0.07dB, so how does these values compared with the common tunable laser? Are they smaller or bigger?

Answer2The linewidth achieved in this paper is better than Ref [4] (0.05nm) [7](0.02nm), and the maximum peak power fluctuation is better than Ref [7] (0.25dB). So by comparing with other reported tunable lasers with the same tuning mechanism, this laser output stability has been greatly improved to our knowledge. The corresponding content is added in line185 marked yellow.

Ref [4]: F. Xiao, K. Alameh, and T. T. Lee, Opt. Express 17, 18676 (2009).

Ref [7]: Q. Ai, X. Chen, M. Tian, B.B. Yan, Y. Zhang, F.J. Song, G. X. Chen, X. Z. Sang, Y. Q. Wang, F. Xiao, and K. Alameh, Appl. Opt. 54, 603 (2015).

Reviewer 2 Report

This is an interesting article that explains an important methodology.  The authors, however, are encouraged to address the issues below:

The letters in fig1 are not defined and explained

use "lens" in fig2a

the relevant literature has not been properly explored. Please see the article : 

https://ieeexplore.ieee.org/abstract/document/6412716

4. What would be the idea DMD for the application explained in this study? what characteristics will make it a dream device?

5.  what are the factors limiting the output power?

Author Response

Reviewer 2:

The reviewer’s comment: This is an interesting article that explains an important methodology. The authors, however, are encouraged to address the issues below:

1. The letters in Fig.1 are not defined and explained.

2. Use "lens" in fig2a

Answer1-2: Thanks for the referee’s kind reminder. We have modified the related errors in Figs.

3. The relevant literature has not been properly explored. Please see the article:

https://ieeexplore.ieee.org/abstract/document/6412716

Answer3: The relevant literature (Ref) investigate the improvement in the nonlinearity of a conventional wavelength swept laser source on the basis of a fiber Fabry-Pérot tunable filter using a well-established optimization method, simulated annealing (SA)  to achieve maximum amplitude for the Fourier transformed peaks of the photodetected interferometric signal. Loading the modulation algorithm on the DMD is an attractive solution, and our research process in the future will also consider using algorithms to further improve the performance of tunable fiber lasers.

Ref: M. R. N. Avanaki, A. Bradu, I. Trifanov, A. B. L. Ribeiro, A. Hojjatoleslami and A. G. Podoleanu, IEEE Photonics Technology Letters, 25(5), 472(2013).

4. What would be the idea DMD for the application explained in this study? What characteristics will make it a dream device?

Answer4: A DMD is an array of thousands of individually addressable and tiltable mirror pixels, manufactured by Very Large Scale Integration (VLSI) technology. These micromirrors driven by voltage can be independently rotated ±q° along the diagonal, corresponding to an “on” or “off” state. By uploading steering holograms onto the DMD controlled by remote software, any waveband of stimulated spontaneous emission spectra can be routed and coupled into the optical system along the original path, and the others are dropped out with dramatic attenuation, thereby realizing the laser selection and wavelength tuning. The tilting micromirrors array causes the diffraction effect, similar with that of a 2D blazed grating. In the tunable fiber laser, the light diffracted by the DMD chip can operate under the near-blazed condition, so that the insertion loss caused by the diffraction is greatly reduced. Besides, the cost of a DMD is lower than that of liquid crystal on silicon (LCoS) in C-band, so a DMD is a dream device applied in our tunable fiber laser structure.

5. What are the factors limiting the output power?

Answer5:

The main factors that limit the laser output power are the following two aspects.

1.      Pumping power of EDFA.

2.      System insertion loss. This loss is mainly due to (1) the diffraction and reflection loss of the DMD; (2) insertion loss of diffraction grating; (3) polarization dependent loss; (4) lens reflection loss and (5) circulator loss; (6) cross dispersion loss. We will continue to further reduce stray light interference and insertion loss by optimized optical path in the future investigation.

Round 2

Reviewer 2 Report

The following text and reference should be added to the article. The relevant literature (Ref) investigate the improvement in the nonlinearity of a conventional wavelength swept laser source on the basis of a fiber Fabry-Pérot tunable filter using a well-established optimization method, simulated annealing (SA)  to achieve maximum amplitude for the Fourier transformed peaks of the photodetected interferometric signal.Loading the modulation algorithm on the DMD is an attractive solution, and our research process in the future will also consider using algorithms to further improve the performance of tunable fiber lasers. Ref: M. R. N. Avanaki, A. Bradu, I. Trifanov, A. B. L. Ribeiro, A. Hojjatoleslami and A. G. Podoleanu, IEEE Photonics Technology Letters, 25(5), 472(2013).

Author Response

Reviewer :

The reviewer’s comment: 

The following text and reference should be added to the article. The relevant literature (Ref) investigate the improvement in the nonlinearity of a conventional wavelength swept laser source on the basis of a fiber Fabry-Pérot tunable filter using a well-established optimization method, simulated annealing (SA)  to achieve maximum amplitude for the Fourier transformed peaks of the photodetected interferometric signal.Loading the modulation algorithm on the DMD is an attractive solution, and our research process in the future will also consider using algorithms to further improve the performance of tunable fiber lasers. Ref: M. R. N. Avanaki, A. Bradu, I. Trifanov, A. B. L. Ribeiro, A. Hojjatoleslami and A. G. Podoleanu, IEEE Photonics Technology Letters, 25(5), 472(2013).

Answer:

Thanks for the referee’s kind advices. The relevant text and reference had been added to the article. The corresponding content is added in line 35, line 195 and line 217 marked yellow.

Round 3

Reviewer 2 Report

The authors have responded to all my comments.